# Polygenic Risk Score (PRS) Combined with NGS Panel Testing Increases Accuracy in Hereditary Breast Cancer Risk Estimation

**DOI:** 10.3390/diagnostics14161826

**Published:** 2024-08-21

**Authors:** Nikolaos Tsoulos, Eirini Papadopoulou, Konstantinos Agiannitopoulos, Dimitrios Grigoriadis, Georgios N. Tsaousis, Dimitra Bouzarelou, Helen Gogas, Theodore Troupis, Vassileios Venizelos, Elena Fountzilas, Maria Theochari, Dimitrios C. Ziogas, Stylianos Giassas, Anna Koumarianou, Athina Christopoulou, George Busby, George Nasioulas, Christos Markopoulos

**Affiliations:** 1Genekor Medical S.A., 15344 Athens, Greece; ntsoulos@genekor.com (N.T.); eirinipapad@genekor.com (E.P.); d.grigoriadis@genekor.com (D.G.); gtsaousis@genekor.com (G.N.T.); d.bouzarelou@genekor.com (D.B.); gnasioulas@genekor.com (G.N.); 2First Department of Internal Medicine, Laikon General Hospital, School of Medicine, National Kapodistrian University of Athens, 11527 Athens, Greece; helgogas@gmail.com (H.G.); ziogasdc@gmail.com (D.C.Z.); 3School of Medicine, National and Kapodistrian University of Athens, 11527 Athens, Greece; ttroupis@med.uoa.gr (T.T.); klinikimastou@yahoo.gr (C.M.); 4Metropolitan Hospital, 18547 Athens, Greece; bennievenizelos@gmail.com; 5Second Department of Medical Oncology, Euromedica General Clinic, 54645 Thessaloniki, Greece; fountzila@oncogenome.gr; 6Oncology Unit, “Hippokrateion” General Hospital of Athens, 11527 Athens, Greece; mtheochari@gmail.com; 7Second Oncology Clinic IASO, General Maternity and Gynecology Clinic, 15123 Athens, Greece; sgiassas@yahoo.com; 8Hematology Oncology Unit, 4th Department of Internal Medicine, School of Medicine, National and Kapodistrian University of Athens, Attikon University Hospital, 12462 Athens, Greece; akoumari@yahoo.com; 9Oncology Unit, ST Andrews General Hospital of Patras, 26332 Patras, Greece; athinachristo@hotmail.com; 10Allelica Inc., 447 Broadway, New York, NY 10013, USA; george@allelica.com

**Keywords:** polygenic risk score (PRS), breast cancer, next-generation sequencing (NGS)

## Abstract

Breast cancer (BC) is the most prominent tumor type among women, accounting for 32% of newly diagnosed cancer cases. BC risk factors include inherited germline pathogenic gene variants and family history of disease. However, the etiology of the disease remains occult in most cases. Therefore, in the absence of high-risk factors, a polygenic basis has been suggested to contribute to susceptibility. This information is utilized to calculate the Polygenic Risk Score (PRS) which is indicative of BC risk. This study aimed to evaluate retrospectively the clinical usefulness of PRS integration in BC risk calculation, utilizing a group of patients who have already been diagnosed with BC. The study comprised 105 breast cancer patients with hereditary genetic analysis results obtained by NGS. The selection included all testing results: high-risk gene-positive, intermediate/low-risk gene-positive, and negative. PRS results were obtained from an external laboratory (Allelica). PRS-based BC risk was computed both with and without considering additional risk factors, including gene status and family history. A significantly different PRS percentile distribution consistent with higher BC risk was observed in our cohort compared to the general population. Higher PRS-based BC risks were detected in younger patients and in those with FH of cancers. Among patients with a pathogenic germline variant detected, reduced PRS values were observed, while the BC risk was mainly determined by a monogenic etiology. Upon comprehensive analysis encompassing FH, gene status, and PRS, it was determined that 41.90% (44/105) of the patients demonstrated an elevated susceptibility for BC. Moreover, 63.63% of the patients with FH of BC and without an inherited pathogenic genetic variant detected showed increased BC risk by incorporating the PRS result. Our results indicate a major utility of PRS calculation in women with FH in the absence of a monogenic etiology detected by NGS. By combining high-risk strategies, such as inherited disease analysis, with low-risk screening strategies, such as FH and PRS, breast cancer risk stratification can be improved. This would facilitate the development of more effective preventive measures and optimize the allocation of healthcare resources.

## 1. Introduction

Breast cancer is the most prominent tumor type among women, accounting for 32% of newly diagnosed cancer cases. Annually, the incidence of this malignancy increases by 0.6% to 1% [1]. As of 2022, the worldwide incidence of breast cancer among women surpassed 2.3 million [2]. The etiology of the disease is frequently unknown, with non-hereditary causes and risk factors predominating; thus, the identification of risk factors that could be utilized for early cancer detection and increased surveillance is critical and continues to be an active area of investigation [3].

Recent research has indicated that a polygenic risk model of inheritance, wherein a substantial quantity of prevalent single-nucleotide polymorphisms (SNPs) collectively contribute to overall risk, may account for certain instances of cancer predisposition [4,5]. The influence of common, low-penetrance genetic loci has been studied using data from genome-wide studies (GWS). Genome-wide association studies use the genetic risk information from the millions of single-nucleotide polymorphisms (SNPs) discovered to determine an individual’s genetic susceptibility to a specific, usually complex, trait. Using this information, the sum of all common, intermediate, and rare variants considered to contribute to disease susceptibility is derived. Interactions within and between these variants form a Polygenic Risk Score (PRS). PRS’s utility has been investigated in several complex diseases as well as several cancers, including breast and prostate cancer [6]. Furthermore, for breast cancer, PRS combined with family history appears to be able to stratify the risk of developing the disease. In addition, studies have shown that PRS analysis can modify the risk in carriers of pathogenic mutations mainly in intermediate-risk genes [7,8,9]. However, there are no studies that indicate the contribution of this analysis to the increase in cancer predisposition in patients with a personal history of the disease.

Multigene analysis by Next-Generation Sequencing (NGS) is a highly effective method for identifying individuals who possess a genetic susceptibility to inherited malignancies, especially in cases involving a familial history of cancers, the occurrence of multiple cancers in a single individual, or the onset of cancer at a younger age [10]. Moreover, a germline alteration in a cancer-associated gene is usually detected in up to 25% of breast cancer patients [11]. However, only 10–15% of them harbor an alteration in a high-risk gene that could explain the disease onset [12]. The remaining cases involve patients with moderate-to-low risk variants, which are not sufficient to explain the development of cancer and therefore cannot be utilized to modify treatment or predict reoccurrence.

Similarly, the identification of a low-risk germline variant in a non-affected individual usually creates confusion even following genetic counselling without providing concrete information about the risk of cancer or the preventive measures required based on the genetic analysis result. Hence, there is a suggestion that PRS could be incorporated into risk models and utilized in such cases to more accurately compute the breast cancer risk and subsequently adjust patients’ stratification into lower- or higher-risk groups, leading to a more individualized risk assessment [4]. Moreover, it could be implemented to identify at-risk patients among the gene-negative group and modify their preventive management accordingly [13].

This study aims to evaluate retrospectively the clinical usefulness of PRS, utilizing a group of patients with a BC diagnosis. The PRS-calculated risk will be incorporated with other risk factors such as the family history of BC and germline variant status. The number of patients that could have been identified by the combination of risk factors will be determined, with an emphasis on the additive value that this multivariable analysis provides for adequate risk prediction. Since our cohort consists of individuals who are already affected, the primary objective is to evaluate the sensitivity of this multi-factor approach in risk assessment.

## 2. Materials and Methods

### 2.1. Patient Information

A total of 105 women with diagnosed breast cancer and hereditary analysis results available were included in the study. The selection was performed in order to include all testing results categories: positive in a high-risk gene, positive in an intermediate/low-risk gene, and negative. Of those, 47 had a family history of breast/ovarian cancer (41 with a first-degree relative with BC), 35 a family history of other tumors, and 23 were without any family history of cancer. The mean age of diagnosis was 47 years (range 27–74) (Table 1).

### 2.2. PRS Calculation

Polygenic Risk Score results were obtained from an external laboratory (Allelica, New York, NY, USA) using DNA genotyping in about 750,000 genetic variants by an Illumina genotyping array, the Global Screening Array (GSA), and Beagle V.5 to impute 50 million additional genome wide variants. The PRS was computed from 577,113 SNPs by summing the number of risk alleles of each SNP weighted by the corresponding effect size. SNPs and effect sizes were retrieved from summary statistics of the GENOME WIDE Association Study (GWAS) on breast cancer as previously reported [14] with an additional selection step using a modified Stacked Clumping and Thresholding approach [15,16]. The PRS model that was chosen has demonstrated substantial stratification potential across a variety of ancestries [17].

Family history was defined as having at least one first-degree relative diagnosed with BC. Individual level risk was computed by first assessing where in a genetic ancestry-matched distribution of PRS values the test individual’s PRS lies, and then converting this percentile into lifetime risk. Risk distributions were built by applying Allelica’s PRS to the UK Biobank dataset [18].

Lifetime risk of breast cancer was estimated using a Cox proportional hazards model of incident breast cancer in the UK Biobank dataset. High risk was defined as a greater than 20% lifetime risk threshold for BC, as recommended by the American Cancer Society [19]. The risks were computed both with and without considering additional risk factors, including gene status and family history. In calculating the risk of BC, pathogenic variants of the *BRCA1/2* and *CHEK2* genes were the only factors taken into account in relation to gene status.

### 2.3. Statistical Analysis

The statistical analysis employed *t*-tests using the ‘ttest_ind’ module from Python’s (version 3.10.9) scipy.stats library [20] to compare groups across different features such as family history, genetic testing outcomes, PRS percentile, PRS risk, and various combined risk estimation strategies. This choice was based on the observation that our data are normally distributed, which is a key requirement for the *t*-test.

Density and scatter plots were generated using Python’s (version 3.10.9) matplotlib [21] and seaborn libraries [22]. We used kernel density estimation (KDE) to visualize the distribution of continuous variables such as PRS percentile and risk scores. Parameters for the kernel density estimation (KDE) figures were adjusted, with the smoothing bandwidth parameter set to ‘Scott’ and the scaling factor (bw_adjust) set to 0.9. The smoothing bandwidth parameter (‘Scott’) and the scaling factor (bw_adjust = 0.9) were chosen to provide a balance between over-smoothing and under-smoothing the data.

## 3. Results

### 3.1. PRS Results

A total of 105 breast cancer (BC) cases that had been genetically analyzed using a capture-based next-generation sequencing (NGS) method to examine 52 cancer-related genes were selected for PRS analysis [23].

The inclusion of all NGS result categories was ensured in the selection of cases that had undergone PRS evaluation. Twenty patients harbored a pathogenic variant in a gene associated with a high risk of developing the disease (*BRCA1*, *BRCA2*, *PALB2*, *TP53*), twenty patients carried a moderate risk variant (*CHEK2*, *ATM*, *BRIP1*, *RAD51C*, *PMS2*), eighteen patients had a pathogenic variant in a gene with low/undetermined correlation with BC, and forty-seven patients had no detectable variation. Whole-Exome Sequencing analysis has also been conducted in all 85 cases without a high-risk gene pathogenic alteration; however, no additional clinically significant alteration was identified.

PRS values for every patient were computed. Patients were categorized into distinct categories according to the presence or absence of family history (FH), the presence of a gene alteration, and the classification of the gene variant as high, moderate, or low risk. The PRS percentile for each case was computed. Cancer risk for each subgroup was determined solely based on the absolute PRS values.

Additionally, risk assessments were computed, incorporating factors such as age, FH, and *BRCA*/*CHEK2* gene status, both with and without PRS, following the provided algorithms.

#### 3.1.1. Percentile of PRS in BC Patients

The utilization of percentile categorization for PRS in our cohort indicates that it has a higher mean PRS than the general population as determined by the UK Biobank (Allelica internal data [24,25] (Figure 1). Moreover, PRS values were increased both in patients with and without a FH of BC. Nevertheless, the more pronounced right-skewed distribution of patients with FH of BC, suggests that the PRS values are even more elevated in this subset of patients (Figure 2).

With respect to the pathogenic variant status, patients with a positive genetic result in a high-risk gene exhibited lower PRS distribution values than those harboring an alteration in a moderate- or low-risk gene. Patients with negative genetic test results appear to have the greatest PRS contribution to breast cancer risk, as indicated by their elevated PRS values (Figure 3). Therefore, there is a statistically significant difference in the PRS percentile observed in patients with a high- or moderate-risk gene alteration compared to patients with negative genetic test results (*p* = 0.013 and *p* = 0.015, respectively). There was no statistically significant difference observed in the PRS values between low-risk gene carriers and non-carriers (*p* = 0.600). Similar observations were obtained when considering the younger patients’ group (<45 years), while no difference in the PRS percentile based on gene status could be observed for older patients (>45 years).

#### 3.1.2. The Contribution of FH and Gene Status in Accurate Risk Estimation

The cancer risk estimation distribution determined by PRS results revealed that an elevated risk of cancer incidence was present in merely 12.38% (13/105) of the cases examined. When considering factors such as FH and gene status, in addition to PRS, it was found that 41.90% (44/105) of the patients who underwent testing exhibited greater than twice the average lifetime risk (Figure 4).

Among the patients with a family history of breast cancer, the implementation of the PRS without considering other risk factors could identify an increased >20% BC risk in 15% of the patients, while this percentage is slightly reduced to 10.77% in patients with no FH. When FH, gene status, and PRS are all considered in the risk estimation, 65.00% of the cases with FH would show an increased risk of BC occurrence, with a mean age of disease diagnosis of 47.31 years (29–67 years). In contrast, this percentage was limited to only 27.69% in patients without a FH of the disease, with a mean age of disease diagnosis of 41.95 (28–64 years) (Figure 5). The difference in the rate of high PRS between women with positive and negative FH of cancer is statistically significant (*p* < 0.01).

Moreover, PRS-calculated risks differ significantly among older and younger patients. In our cohort, 16.42% (11/67) of the patients <45 showed increased PRS-calculated BC risk (median risk 15.5%), compared to solely 5.26% (2/38) of those ≥45 (median risk 11.5%).

Furthermore, PRS results indicative of a high risk of BC were identified more frequently in the 47 patients lacking a gene alteration, comprising 25.53% of the cases. The median PRS-based calculated risk of BC was 16%, which was further empowered if the FH information was incorporated in the risk calculation (18.1%). In contrast, only 1.72% of the gene-positive cases would have shown an increased BC risk based on the PRS results if the genetic testing analysis had not been performed. A low penetrance *CHEK2* variation was present in the only patient among the gene-positive cohort with an increased BC risk of 20.4% based solely on PRS (or 28.2% when the FH of BC in the family is considered). The difference in the rate of high PRS between women who tested positive for pathogenic mutations and those with negative genetic testing results is also statistically significant (*p* < 0.01) (Figure 6). In addition, the PRS-based risk values were considerably lower, with a median PRS-based risk of 12.15%. This value is very close to the population risk of breast cancer, as determined by the Surveillance, Epidemiology, and End Results (SEER) registry, which is 13% [1]. As a result, we could deduce that the BC incidence in gene-positive cases is not of polygenic origin but probably has a monogenic etiology, thereby elevating the significance of single-gene involvement in such instances. In the gene-negative patients, though, the polygenic etiology of cancer is more plausible, and it leads to increased PRS values.

## 4. Discussion

Genetic testing is considered mandatory for women with FH of BC or a personal history of BC and/or specific clinical features [26]. Even though such an analysis can determine the risk of disease in cases with a pathogenic finding, it is not informative concerning the BC risk occurrence in case of a negative result.

PRS is increasingly used to screen the general population for the estimation of several disease risk assessments, including cancer. Therefore, the integration of PRS into routine healthcare decision-making has been proposed in several studies [27,28].

Even though this approach could be useful to identify individuals with high PRS and therefore increased need of surveillance, it could also lead to complacency in case of negative PRS results. Therefore, the identification of the appropriate population to be screened is of great utility.

### 4.1. PRS-Based BC Risk Estimations

In this study we retrospectively evaluated the risk estimations obtained by PRS in patients already diagnosed with breast cancer, taking also into account other important factors such as breast cancer FH and NGS-based genetic test results for inherited alterations in BC-related genes. Given that BC has already manifested in our cohort, the objective of this study was to examine whether adding PRS to other risk factors could enhance the detection rate of high-risk individuals who would have benefited from preventive screening prior to the disease’s onset. Our results indicate that the addition of PRS in cancer risk evaluation significantly increases the sensitivity of such risk estimation.

The gene-negative group exhibited a higher prevalence of high PRS detection, accounting for 25.53% of the cases, in contrast to the gene-positive group’s 1.72%. The disparity in PRS values between women who received negative genetic testing results and those who tested positive for a pathogenic alteration was identified as statistically significant (*p* = 0.0028). Hence, PRS appears to exert a negligible influence on risk estimation in individuals carrying high or intermediate risk alleles, whereas the monogenic variant represents the most likely cause of BC. In negative patients, the greatest influence on disease occurrence is attributed to polygenic etiology.

In the population of FH-positive patients, the combination of PRS, FH, and gene status would have identified 65% of high-risk cases. In FH-negative cases, 27.69% of patients at increased risk would be detected, indicating a less significant contribution of PRS analysis.

The major utility was also seen in FH-positive patients with negative genetic test results where the positivity in terms of increased risk of cancer development reached the percentage of 63.63% (14/22) patients Therefore, PRS’s utility seems to be increased in FH-positive cases, identifying a bigger percentage of at-risk cases.

These findings align with a prior investigation that examined the efficacy of PRS when combined with additional risk factors, including family history and positivity in a gene associated with moderate to high susceptibility to disease. A combination of risk factors enhanced the positive predictive value by 45 to 50%, while a high PRS identified 39.5% of the women who were more likely to receive a breast cancer diagnosis [29].

Moreover, a study investigating a genetic risk-based approach in identifying women at low risk of disease has shown that BC incidence was significantly reduced in the absence of variant *BRCA1*, *BRCA2*, *PALB2*, *ATM*, or *CHEK2* genes and a low polygenic risk score (hazard ratio, 0.39) [30].

### 4.2. Clinical Applicability of PRS-Guided BC Risk Assessment

Notwithstanding the critiques pertaining to the clinical applicability of PRS-guided risk assessment, its integration with additional risk factors can yield substantial advantages in our study [31,32]. The implementation of PRS among women with a FH of BC in our cohort contributed substantially to the detection of a greater proportion of at-risk cases within patients with negative genetic testing results. Among these patients, FH combined with PRS could indicate increased cancer risk in more than 63% of the cases, highlighting the need for enhanced preventive screening strategies to prevent the onset of cancer in this subset. These results are in accordance with previous studies showing that PRSs could explain 10–18% of familial breast cancer risk [33,34].

However, when FH is absent, the proportion of high-risk detections among patients with NGS-negative status is diminished to 28%. Considering that the individuals in our cohorts have already suffered from BC, we could deduce that in these cases PRS analysis would have failed to identify at-risk individuals in 70% of the cases, showing a lack of sensitivity in cancer risk detection in the absence of other risk factors.

Although PRS values are higher in the patients with younger disease onset compared with those who were diagnosed with cancer at >45 years, the utility of PRS in detecting cancer risk seems constant in the gene-negative cohort independently of age, with 37.5% and 52.17% high BC risk, respectively. The addition of the family history parameter is more meaningful in the younger patients though increasing the rate of higher-risk patients to 75% (6/8), compared to 57.14% (8/14) in the older group.

Our results indicate that although PRS tends to be higher in our cohort than in the general population, it cannot replace other risk factors such as FH and genetic analysis results. However, it can be used be used in combination with them to increase the rate of at-risk individuals who require enhanced surveillance to prevent disease or for early disease detection. Despite these results, the implementation of all risk factors still misses more than half of the women who eventually developed BC in our cohort. Therefore, attention should be given to the appropriate interpretation of the results obtained when performing such analysis paying special attention to explicitly explaining that a negative result does not exclude the possibility of cancer and should not lead to complacency and diminished surveillance may result inadvertently.

Numerous studies have demonstrated the superiority of PRS implementation in enhancing population estimates of women’s breast cancer risk in comparison to the use of family history or pathogenic variants alone [30]. Research has shown that integrating PRS into breast cancer risk assessment can lead to cost-effective risk stratification strategies [35,36]. However, the extent to which polygenic risk stratification contributes to cost-effective cancer screening remains uncertain [37]. Based on these results, it is crucial to incorporate PRS into population-wide initiatives to improve the precision of breast cancer risk assessment among study participants. Nevertheless, the implementation of PRS in clinical settings requires the consideration of several critical factors. These include the absence of evidence-based guidelines, the lack of standardization in test methodology and reporting, and the absence of patient education to facilitate comprehension of the results [38].

To enhance the applicability of PRS in clinical settings for the evaluation of breast cancer risk and personalized management, it is essential to assess a variety of components. The development of robust decision support tools for healthcare providers, the establishment of effective monitoring and evaluation mechanisms, the evaluation of the costs associated with PRS testing, the provision of comprehensive patient education on the implications and limitations of PRS results, and the assurance of accessibility to testing facilities are among the actions to consider.

To address obstacles, guarantee equitable access to PRS testing, and optimize the advantages of personalized breast cancer care, healthcare providers, researchers, policymakers, genetic counsellors, and patient advocacy groups must collaborate [39].

### 4.3. Limitations of the Study

The most significant limitation of the study is the number of patients available for PRS calculation. However, these patients were derived from a breast cancer patient population with known genetic test results and included both individuals with and without a family history of cancer. Unfortunately, other clinical data were not available.

Another limitation is that the lifetime risk is estimated from the PRS alone using a model based on the relative risk conferred by the PRS and a baseline average lifetime risk of 12%. Other risk estimation models such as the CanRisk/BOADICEA model were not included in the analysis, partially because this model is designed to estimate future risks of developing cancer, while the population studied has already developed the disease [40,41].

Additionally, the retrospective design of the study restricts its ability to draw definitive conclusions regarding the potential of PRS integration in clinical practice.

## 5. Conclusions

To our knowledge, this is the first study in a Greek population to calculate the PRS in women with breast cancer while also considering FH and the outcomes derived from NGS analysis for inherited cancer syndromes. Our results indicate a major utility of PRS calculation in women with FH in the absence of a monogenic etiology detected by NGS. By combining high-risk strategies, such as inherited disease analysis, with low-risk screening strategies, such as FH and PRS, breast cancer risk stratification can be improved. This would facilitate the development of more effective preventive measures and optimize the allocation of healthcare resources. However, it is important to note that the current evidence supporting the use of PRS in clinical practice is still limited. Consequently, although PRS has the potential to enhance personalized medicine and risk prediction, additional research is required to gain a comprehensive understanding of its practical application and impact on clinical practice.

## Figures and Tables

**Figure 1 diagnostics-14-01826-f001:**
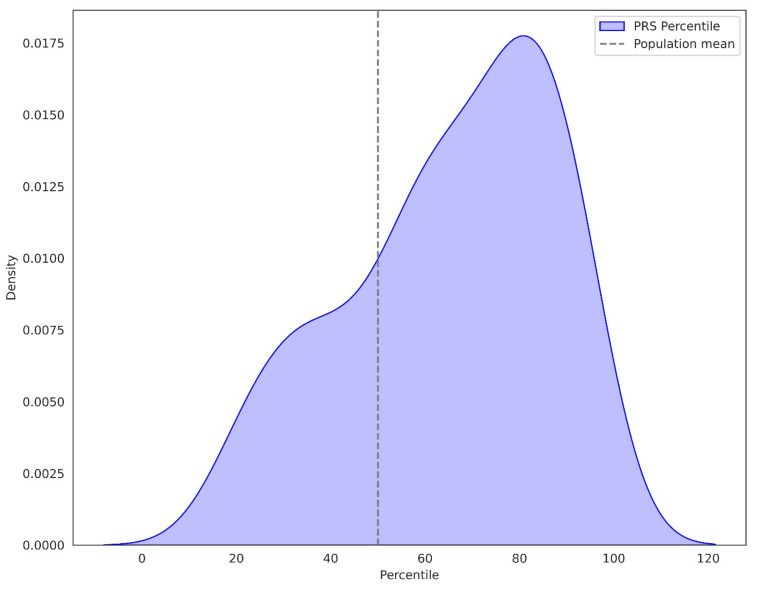
PRS percentile distribution in 105 patients with breast cancer. The dotted line represents the average PRS expected in the general population.

**Figure 2 diagnostics-14-01826-f002:**
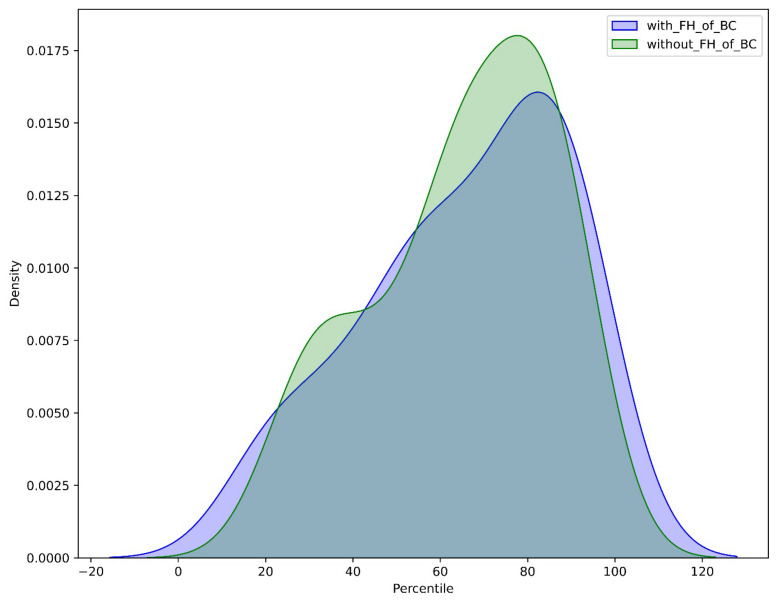
PRS percentile distribution in patients with and without family history (FH) of breast cancer. Blue: PRS-based risk estimation in patients with FH of BC. Green: PRS-based risk estimation in patients without FH of BC.

**Figure 3 diagnostics-14-01826-f003:**
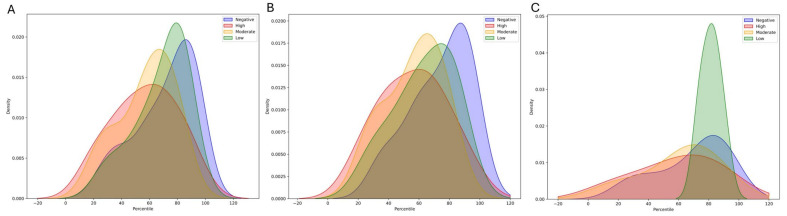
(**A**) PRS percentile distribution in patients without pathogenic gene alterations and with alterations in high-, moderate-, and low/unspecified-risk genes. (**B**) PRS percentile distribution in patients <45 years without pathogenic gene alterations and with alterations in high-, moderate-, and low/unspecified-risk genes. (**C**) PRS percentile distribution in patients >45 years without pathogenic gene alterations and with alterations in high-, moderate-, and low/unspecified-risk genes. Blue: PRS-based risk estimation in patients without a cancer gene alteration detected by NGS. Green: PRS-based risk estimation in patients with a variant in a low-risk gene. Orange: PRS-based risk estimation in patients with a variant in a moderate-risk gene. Red: PRS-based risk estimation in patients with a variant in a high-risk gene.

**Figure 4 diagnostics-14-01826-f004:**
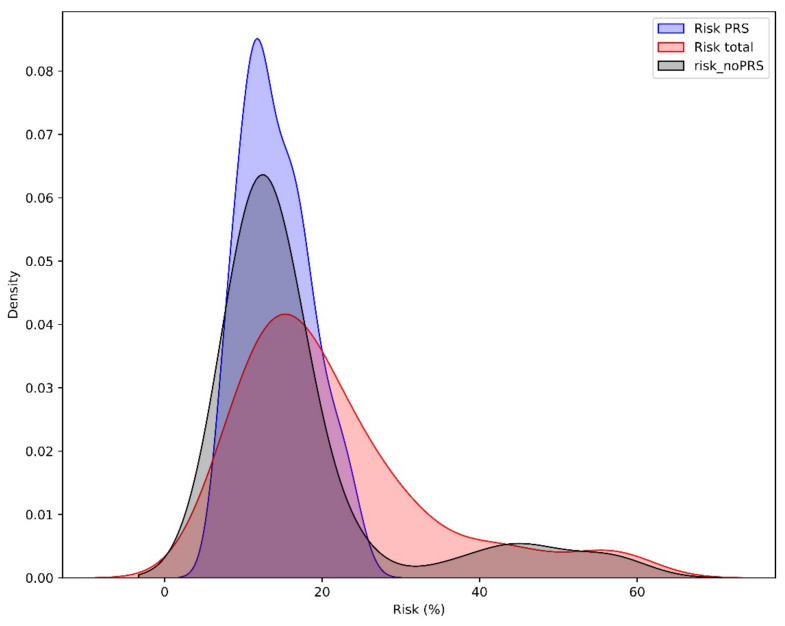
Density curves of BC risk distribution in the cohort of BC patients. Red density curve: BC risk estimation based on PRS only (Risk PRS), Blue: BC risk estimation based on FH and gene status without PRS (risk_noPRS). Grey: BC risk estimation including all factors in addition to PRS (Risk total).

**Figure 5 diagnostics-14-01826-f005:**
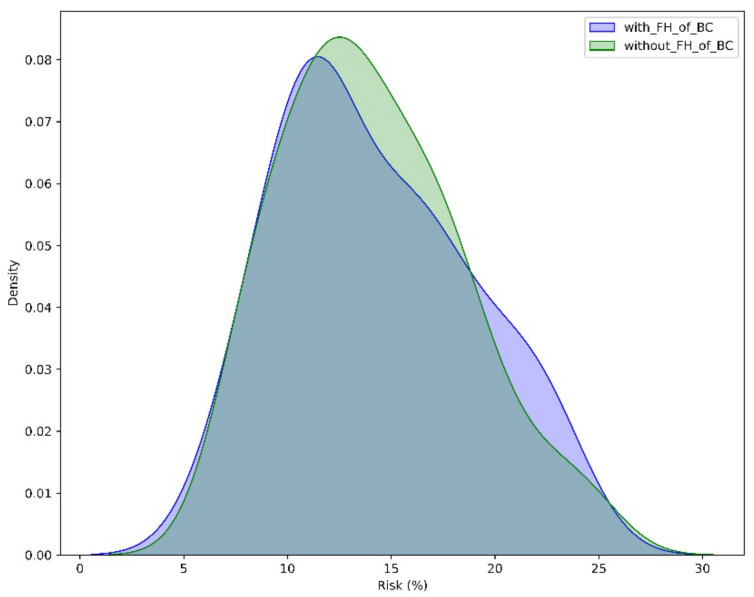
Density curves of PRS-based risk estimation in patients with and without FH of BC. Blue: PRS-based risk estimation in patients with FH of BC. Green: PRS-based risk estimation in patients without FH of BC.

**Figure 6 diagnostics-14-01826-f006:**
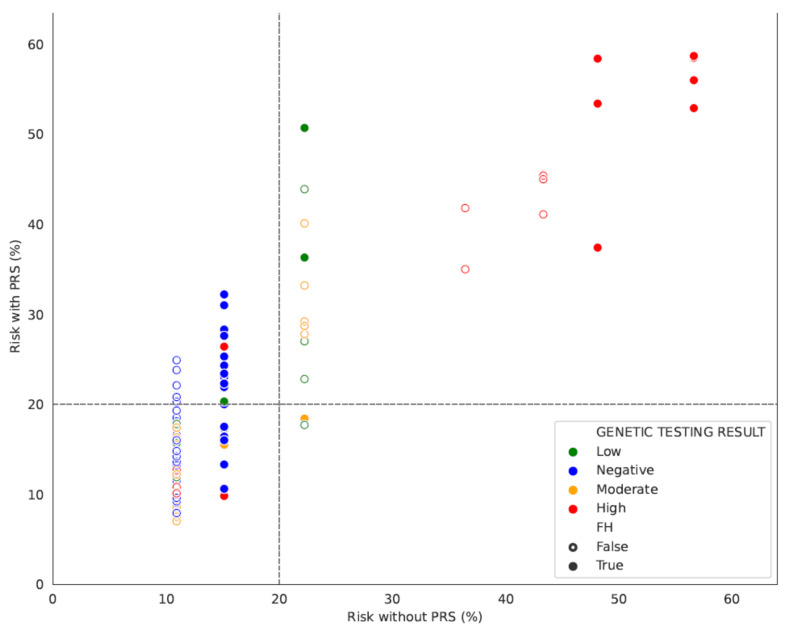
Risk distribution with and without the integration of PRS results in patients categorized according to their FH/gene status.

**Table 1 diagnostics-14-01826-t001:** Cohort characteristics that were incorporated into the study.

	All	With FH of BC	Without FH of BC
Age at Diagnosis			
Number of Women < 45 yo	67	20	47
Number of Women > 45 yo	38	20	18
Average age	47	47.6	42.7
Genetic testing			
Number of Negative cases	47	22	25
Number of Pathogenic in high risk gene	20	11	9
Number of Pathogenic in moderate risk gene	20	3	17
Number of Pathogenic in low/unspesified risk gene	18	4	14

## Data Availability

Data is contained within the article.

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
