# Peer review of "Polygenic Risk Score (PRS) Combined with NGS Panel Testing Increases Accuracy in Hereditary Breast Cancer Risk Estimation"

_diagnostics, 2024, doi:10.3390/diagnostics14161826_

Round 1
Reviewer 1 Report
Comments and Suggestions for Authors
Dear Authors,
The manuscript titled "Polygenic Risk Score (PRS) Combined with NGS Panel Testing Increases Accuracy in Hereditary Breast Cancer Risk Estimation" presents significant contributions to the field of hereditary breast cancer risk assessment. But several concerns need to be addressed to improve the manuscript's quality and clarity:
This study is the first of its kind in a Greek population to integrate PRS with family history and NGS results, addressing a crucial gap in breast cancer risk stratification. This adds substantial value to the current understanding and provides a model that could be replicated in other populations.
The inclusion of multiple risk factors such as FH, genetic variants, and PRS in the analysis is commendable. This comprehensive approach strengthens the validity of the findings and highlights the complex interplay between these factors in breast cancer risk.
The use of advanced statistical methods and robust data analysis techniques, including t-tests and kernel density estimation, ensures the reliability and accuracy of the results. The detailed explanation of the PRS calculation and risk estimation processes is also appreciated.
The results provide compelling evidence that combining PRS with FH and NGS results improves the identification of high-risk individuals. The finding that PRS is particularly useful in FH-positive, gene-negative patients is a critical insight for clinical practice.
The implications of the study for preventive strategies and healthcare resource allocation are well-articulated. The work underscores the importance of integrating various risk assessment tools to enhance early detection and prevention of breast cancer.
The manuscript is, at times, difficult to follow due to the dense presentation of information. Improving the organization and flow of the sections, particularly the Results and Discussion, would enhance readability and comprehension. Consider breaking down complex paragraphs and using subheadings to guide the reader.
While the inclusion of 105 breast cancer patients is a start, the sample size may be too small to draw broad conclusions. Additionally, more detailed demographic information about the cohort would provide context and allow readers to assess the generalizability of the findings.
Although several statistical methods were used, the justification for choosing specific tests and the interpretation of p-values need further clarification. For instance, why were certain statistical thresholds chosen, and how do they impact the study's conclusions?
The PRS calculation method, while described, could benefit from additional validation against independent datasets. Providing evidence of the PRS model's performance in different populations would strengthen the reliability of the findings.
The manuscript lacks a thorough discussion of the study's limitations. Addressing potential biases, the limitations of the retrospective design, and the challenges of integrating PRS into clinical practice would provide a more balanced perspective.
While the potential of PRS in risk stratification is discussed, there is a need for a more detailed exploration of how PRS could be practically implemented in clinical settings. This includes considerations of cost, accessibility, and patient education.
The conclusion, while optimistic, may overstate the current applicability of PRS in clinical practice. It is important to temper the conclusions with a realistic appraisal of the current evidence and the need for further research.
Author Response
Dear Reviewer
Thank you for your valuable contribution and constructive suggestions. Please find below the responses to your comments.
Comment 1: The manuscript is, at times, difficult to follow due to the dense presentation of information. Improving the organization and flow of the sections, particularly the Results and Discussion, would enhance readability and comprehension. Consider breaking down complex paragraphs and using subheadings to guide the reader.
Response 1: Entitled subsections are present in the results sections and are also added in the discussion section
Comment 2: While the inclusion of 105 breast cancer patients is a start, the sample size may be too small to draw broad conclusions. Additionally, more detailed demographic information about the cohort would provide context and allow readers to assess the generalizability of the findings.
Response 2: Thank you for your comment
The following table was added in the manuscript:
Table1: Cohort characteristics that were incorporated into the study
|
|
All |
With FH |
Without FH |
|
Age at Diagnosis |
|||
|
Number of Women <45 yo |
67 |
20 |
47 |
|
Number of Women >45 yo |
38 |
20 |
18 |
|
Average age |
47 |
47.6 |
42.7 |
|
Genetic testing |
|||
|
Number of Negative cases |
47 |
22 |
25 |
|
Number of Pathogenic in high risk gene |
20 |
11 |
9 |
|
Number of Pathogenic in moderate risk gene |
20 |
3 |
17 |
|
Number of Pathogenic in low/unspesified risk gene |
18 |
4 |
14 |
Comment 3: Although several statistical methods were used, the justification for choosing specific tests and the interpretation of p-values need further clarification. For instance, why were certain statistical thresholds chosen, and how do they impact the study's conclusions?
Response 3: The statistical analysis section was rephrased to: The statistical analysis employed t-tests using the 'ttest_ind' module from Py-thon's scipy.stats library [18] to compare groups across different features such as fam-ily history, genetic testing outcomes, PRS percentile, PRS risk, and various combined risk estimation strategies. This choice was based on the observation that our data is normally distributed, which is a key requirement for the t-test.
Density and scatter plots were generated using Python's matplotlib [19] and sea-born libraries [20]. We used kernel density estimation (KDE) to visualize the distribu-tion of continuous variables such as PRS percentile and risk scores. Parameters for the kernel density estimation (KDE) figures were adjusted, with the smoothing bandwidth parameter set to 'Scott' and the scaling factor (bw_adjust) set to 0.9. The smoothing bandwidth parameter ('Scott') and the scaling factor (bw_adjust=0.9) were chosen to provide a balance between over-smoothing and under-smoothing the data
Comment 4: The PRS calculation method, while described, could benefit from additional validation against independent datasets. Providing evidence of the PRS model's performance in different populations would strengthen the reliability of the findings.
Response 4: The model used has demonstrated substantial stratification potential across a variety of ancestries https://www.researchsquare.com/article/rs-4022359/v1
Comment 5: The manuscript lacks a thorough discussion of the study's limitations. Addressing potential biases, the limitations of the retrospective design, and the challenges of integrating PRS into clinical practice would provide a more balanced perspective.
Response 5: Limitations of the study added :
The most significant limitation of the study is the number of patients available for PRS calculation. However, these patients were derived from a Breast Cancer patient population with known genetic test results and included both individuals with and without family history of cancer.
Another limitation is that the lifetime risk is estimated from the PRS alone using a model based on the relative risk conferred by the PRS and a baseline average lifetime risk of 12%. Other risk estimation models such as CanRisk/BOADICEA model were not included in the analysis, partially because this model is designed to estimate future risks of developing cancer, while the population studied has already developed the disease [33, 34].
Additionally, the retrospective design of the study restricts its ability to draw definitive conclusions regarding the potential of PRS integration in clinical practice.
Comment 6: While the potential of PRS in risk stratification is discussed, there is a need for a more detailed exploration of how PRS could be practically implemented in clinical settings. This includes considerations of cost, accessibility, and patient education.
Response 6: The following paragraphs were added: Numerous studies have demonstrated the superiority of PRS implementation in enhancing population estimates of women's breast cancer risk in comparison to the use of family history or pathogenic variants alone [28]. Research has shown that integrating PRS into breast cancer risk assessment can lead to cost-effective risk stratification strategies [35,36]. However, the extent to which polygenic risk stratification contributes to cost-effective cancer screening remains uncertain (37). Based on these results, it is crucial to incorporate PRS into population-wide initiatives to improve the precision of breast cancer risk assessment among study participants. Nevertheless, the implementation of PRS in clinical settings requires the consideration of several critical factors. These include the absence of evidence-based guidelines, the lack of standardization in test methodology and reporting, and the absence of patient education to facilitate comprehension of the results [38].
To enhance the applicability of PRS in clinical settings for the evaluation of breast cancer risk and personalized management, it is essential to assess a variety of components. The development of robust decision support tools for healthcare providers, the establishment of effective monitoring and evaluation mechanisms, the evaluation of the costs associated with PRS testing, the provision of comprehensive patient education on the implications and limitations of PRS results, and the assurance of accessibility to testing facilities, are among the actions to consider.
To address obstacles, guarantee equitable access to PRS testing, and optimize the advantages of personalized breast cancer care, healthcare providers, researchers, policymakers, genetic counsellors, and patient advocacy groups must collaborate (39).
Comment 7: The conclusion, while optimistic, may overstate the current applicability of PRS in clinical practice. It is important to temper the conclusions with a realistic appraisal of the current evidence and the need for further research.
Response 7: The following phrases were added in the conclusion section: However, it is important to note that the current evidence supporting the use of PRS in clinical practice is still limited. Consequently, although PRS have the potential to enhance personalized medicine and risk prediction, additional research is required to gain a comprehensive understanding of its practical application and impact on clinical practice.
Reviewer 2 Report
Comments and Suggestions for Authors
Abstract:
Abstract far too long, try to focus on main results or provide an additional short abstract; Methods are missing in the abstract
Introduction/Biblography:
Relevant citations missing on BC-PRS, PRS risk estimation (reviews are cited (5), better to use original papers, PMID: 34101481, PMID: 36137616
Methods:
CAVE! Cohort is really small
Inclusion criteria very broad, age 27-74, analysis in subgroups needed but subgroups will remain to small
Which PRS modell was used? 313 (Mavaddat; PMID: 36649146), 306 (BCAC)? Please provide information!
Usually, 10-year risk was used, not life time risk. Most people use CanRisk vor Risk estimation based on BOADICEA (PMID: 36137616). Can you please comment on this? https://www.canrisk.org/
Did the authors correct their PRS values for the greeks mean PRS value? Please check PMID: 38410445
Results:
3.1.1. because of the broad age, please re-analyse in age related subgroups, also for figure 3
"3.1.2. When considering factors such as FH, gene status, and PRS, it was found that 41.90% (44/105) of the patients who underwent testing exhibited a PRS value that was greater than twice the average lifetime risk (Figure 4)." This sentence is missleading. The elevated PRS leads to a increased life time risk!
Line 193: true and false FH? positive or negative FH instead (also in Fig 6)
Figure 4 and 5: please make figure legend more clear. Maybe box plots will make results more easy to get
Discussion:
The authors want to use their results for healthcare decision. They set a treshold of 20% increased lifetime risk but is this the treshold in the greek healthcare system to include or not include patients to surveillance programs? Can you calculate the amount of persons for which surveillance would change after incorporation in the risk calculation? as they did: PMID: 36137616
Major weakness: Non-genetic factors, which are know to have big impact, are not taken into account. " In negative patients, the greatest influence on disease 260 occurrence is attributed to polygenic etiology."
Comments on the Quality of English LanguageQuality of English is ok, however medical and genetic terms can be rephrased
Author Response
Dear Reviewer
Thank you for your valuable contribution and constructive suggestions. Please find below the responses to your comments.
Comment 1: Abstract: Abstract far too long, try to focus on main results or provide an additional short abstract; Methods are missing in the abstract
Response 1: The abstract was modified based on the reviewer’s suggestions and based on the format. The methodology is mentioned but no separate section is permitted.
Comment 2: Introduction/Biblography: Relevant citations missing on BC-PRS, PRS risk estimation (reviews are cited (5), better to use original papers, PMID: 34101481, PMID: 36137616
Response 2: References were added
Comment 3: Methods: CAVE! Cohort is really small
Response 3: The population used was indeed small but balanced to include patients with alterations in high, moderate and low-risk genes as well as negative cases. Moreover it included a balanced number of individuals with and without FH. Therefore, we believe it is representative and can be used to draw important conclusions concerning the utility of the PRS analysis. This is also mentioned in the limitations of the study. Moreover, WES analysis was also performed in the cases without high-risk gene alterations identified (sentence added in p.4, line 160: Whole Exome Sequencing analysis has also been conducted in all 85 cases without a high-risk gene pathogenic alteration; however, no additional clinically significant alteration was identified. ). The aim was to better evaluate the monogenic factors in BC risk estimation in these cases.
Comment 4: Inclusion criteria very broad, age 27-74, analysis in subgroups needed but subgroups will remain to small
Response 4: The inclusion criteria were the presence of a genetic result available and the information concerning FH of breast cancer.
Comment 5: Which PRS modell was used? 313 (Mavaddat; PMID: 36649146), 306 (BCAC)? Please provide information!
Response 5: We used the PRS described in the article https://www.researchsquare.com/article/rs-4022359/v1
Comment 6: Usually, 10-year risk was used, not life time risk. Most people use CanRisk vor Risk estimation based on BOADICEA (PMID: 36137616). Can you please comment on this? https://www.canrisk.org/
Response 6: we did not estimate risk using CanRisk/BOADICEA -- the lifetime risk is estimated from the PRS alone using a model based on the relative risk conferred by the PRS and a baseline average lifetime risk of 12%
Comment 7: Did the authors correct their PRS values for the greeks mean PRS value? Please check PMID: 38410445
Response 7: We did not correct for greek means/sd. We used a principal component based adjustment of the raw PRS based on a global dataset (1000 Genomes Project; described in Busby et al 2023 Nature Communications) and used a European specific reference dataset to assess the overall risk conferred by the PRS.
Comment 8: 3.1.1. because of the broad age, please re-analyse in age related subgroups, also for figure 3
Response 8: Figure 3 was modified accordingly, and the following sentence was added: Similar observations were obtained when considering the younger patients’ group <45 years, while no difference in the PRS percentile based on gene status could be observed for elder patients (>45 years).
Comment 9: "3.1.2. When considering factors such as FH, gene status, and PRS, it was found that 41.90% (44/105) of the patients who underwent testing exhibited a PRS value that was greater than twice the average lifetime risk (Figure 4)." This sentence is missleading. The elevated PRS leads to a increased life time risk!
Response 9: Rephrased to: When considering factors such as FH and gene status, in addition to PRS, it was found that 41.90% (44/105) of the patients who underwent testing exhibited greater than twice the average lifetime risk (Figure 4).
Comment 10: Line 193: true and false FH? positive or negative FH instead (also in Fig 6)
Response 10: Rephrased
Comment 11: Figure 4 and 5: please make figure legend more clear. Maybe box plots will make results more easy to get
Response 11: Legends changed to:
Figure 4. Density curves of BC risk distribution in the cohort of BC patients. Red Density curve: BC risk estimation based on PRS only (Risk PRS). Blue: BC risk estimation based on FH and gene status without PRS (risk no PRS). Grey: BC risk estimation including all factors on addition to PRS (Risk total).
Figure 5. Density curves of PRS-based risk estimation in patients with and without FH of BC. Blue: PRS-based risk estimation in patients with FH of BC. Green: PRS-based risk estimation in patients without FH of BC
Comment 12: The authors want to use their results for healthcare decision. They set a treshold of 20% increased lifetime risk but is this the treshold in the greek healthcare system to include or not include patients to surveillance programs? Can you calculate the amount of persons for which surveillance would change after incorporation in the risk calculation? as they did: PMID: 36137616
Response 12: In the Greek healthcare system, there are no established factors to include or not include patients to surveillance programs.
Comment 13: Major weakness: Non-genetic factors, which are know to have big impact, are not taken into account. " In negative patients, the greatest influence on disease 260 occurrence is attributed to polygenic etiology."
Response 13: Unfortunately, other factors were not available for the individuals tested.
Best Regards